# Grading Systems for Canine Urothelial Carcinoma of the Bladder: A Comparative Overview

**DOI:** 10.3390/ani12111455

**Published:** 2022-06-04

**Authors:** Eleonora Brambilla, Veronica M. Govoni, Alexandre Matheus Baesso Cavalca, Renée Laufer-Amorim, Carlos Eduardo Fonseca-Alves, Valeria Grieco

**Affiliations:** 1Department of Veterinary Medicine and Animal Science, University of Milan, Via dell’ Università 6, 26900 Lodi, Italy; eleonora.brambilla@unimi.it; 2Department of Veterinary Surgery and Animal Reproduction, Sao Paulo State University-UNESP, Botucatu 18618681, Brazil; veronica.m.govoni@unesp.br (V.M.G.); a.cavalca@unesp.br (A.M.B.C.); carlos.e.alves@unesp.br (C.E.F.-A.); 3Department of Veterinary Clinic, Sao Paulo State University-UNESP, Botucatu 18618681, Brazil; renee.laufer-amorim@unesp.br

**Keywords:** urothelial carcinoma, urinary bladder, dog, BUC, grading system

## Abstract

**Simple Summary:**

Tumor histological grading systems are a tool widely used by human pathologists in oncology to support the assessment of tumor behavior and patient prognosis by clinical oncologists. In veterinary medicine, several tumor types already have a histological grading system used for these purposes, but some of these schemes lack reproducibility or correlation with clinical parameters, such as the correlation of the grade with survival time. This is the case for the grading systems proposed for canine bladder urothelial carcinoma. Over the years, some grading systems have been described for this type of tumor in dogs but without any routine use by pathologists and, consequently, without any application in clinical practice either. Based on this fact, the present study aimed to review the histological grading systems that exist in both human and veterinary medicine for bladder urothelial carcinoma, carrying out a critical analysis of their differences and thereby encouraging their real practical use and application in a relevant number of cases, prospectively. In this way, a histological grading system could be chosen or built from the existing ones and the knowledge about the behavior of this neoplasm in canine species could be improved, helping clinicians to establish a prognosis and personalized treatment for each patient with bladder urothelial carcinoma and also consider the predictive markers associated with treatment outcomes.

**Abstract:**

The relationship between tumor morphology and clinical behavior is a key point in oncology. In this scenario, pathologists and clinicians play a pivotal role in the identification and testing of reliable grading systems based on standardized parameters to predict patient prognosis. Dogs with bladder urothelial carcinoma (BUC) were recently proposed as a “large animal” model for the study of human BUCs due to the similar morphology and metastasis locations. BUC grading systems are consolidated in human medicine, while in veterinary medicine, the BUC grading systems that have been proposed for canine tumors are not yet applied in routine diagnostics. These latter systems have been proposed, decade by decade, over the last thirty years, and the reason for their scarce application is mainly related to a lack of specific cutoff values and studies assessing their prognostic relevance. However, for any prognostic study, reliable grading is necessary. The aim of the present article was to give an overview of the BUC grading systems available in both human and veterinary pathology and provide an extensive description and a critical evaluation to support veterinary researchers in the choice of possible grading systems to apply in future studies on canine BUCs.

## 1. Introduction

The relationship between tumor morphology and clinical behavior is a key point in oncology and, in this scenario, pathologists and clinicians play a pivotal role in the identification and testing of reliable grading systems useful for patient prognosis and predicting treatments. The term “tumor grading” refers to the microscopic assessment and quantification of the parameters correlated with the putative clinical aggressiveness of a neoplasm based on the tumor’s histomorphology [1].

The relationship between tumor morphology and the clinical behavior of tumors has been known since the early studies of Rudolf Virchow (1821–1902); however, the first attempts to correlate the microscopic features of tumors with their biology and clinical behavior are traditionally attributed to David Paul von Hansemann (1858–1920) [2,3,4], who, in 1880, systematically studied the microscopic pathology of tumors. Then, in the 1920s, Albert C. Broders and colleagues proposed a grading system for squamous cell carcinoma of the lip and skin, correlating the histologic grade with a patient’s clinical outcome [5,6]. The grading of tumors was subsequently adopted by other pathologists and applied to tumors in other organs.

However, some of the created grading systems are unwieldy, unreliable, and not always reproducible [7]. An ideal system should be simple, easy to apply, reproducible, and useful in clinical practice [7]. In both human and veterinary medicine, with the increase in the number of treatment options, efficient grading systems have become a necessity for classifying patients according to the biological behavior of their tumor [4]. The old systems have been reviewed and improved using advanced techniques and can reduce interobserver variability, improve reproducibility, and determine reliable correlations between treatments and outcomes. Currently, tumor grading assessment varies according to tumor type, and in some instances, more than one grading system is available for some tumors, and two-, three-, or four-tier grading systems are used [1,7]. In veterinary medicine, there is an increasing interest in grading systems that have generally been developed from human tumors and adapted to animal tumors or have been formulated specifically for veterinary medicine.

Bladder urothelial carcinoma (BUC) is an important human disease worldwide, with more than 400,000 new cases per year [8,9]. Dogs with invasive BUCs were recently proposed as a “large animal” model for the study of their human counterparts because they show similar morphology and metastasis locations [10]. Moreover, BUCs comprise 1.5–2% of all naturally occurring cancers in dogs—a rate similar to that reported in humans [11]. BUC tumor grading systems have been available for decades and widely applied in human medicine, while in veterinary medicine, the BUC grading systems proposed for canine tumors are not yet applied in routine diagnostics. The available canine BUC grading systems were proposed by Valli (1995), Patrick (2006), and Meuten (2017) based on cell morphology and infiltration. However, they are poorly applied, possibly for various reasons: their unknown relevance for prognosis and therapy, the late stage of the tumor at the time of diagnosis in the vast majority of dogs, and limited acceptance among pathologists in adopting new grading systems, among others [12,13,14].

Therefore, the aim of the present article was to provide an overview of the BUC grading systems available in both human and veterinary pathology and provide an extensive description and a critical evaluation to support veterinary researchers in the choice of possible grading systems to use in future studies on canine BUCs.

## 2. Canine BUCs: Histological Description

The definitive diagnosis of BUCs requires the histopathologic examination of tissue samples obtained by cystotomy, cystoscopy, or urethral catheterization (cytology) [15]. For the optimal management of BUCs, an extensive pathological description is required, which should include cell morphology, tumor architecture, grade, depth of invasion, tumor differentiation (urothelial or divergent), and tumor stroma (including presence and extent of inflammation) [12,14].

In veterinary medicine, the World Health Organization’s (WHO) 2004 classification of domestic animal tumors is in use, which divides tumors in two categories: papillary or nonpapillary and invasive or noninvasive. Invasion can be limited to the bladder lamina propria or can involve the muscle layer with, in some cases, an extension into the serosa [16].

BUCs are mainly diagnosed in dogs and cattle, while they are rare in other domestic species such as cats and horses [14]. BUC is the most common type of urinary bladder cancer in dogs, affecting 10,000 dogs worldwide each year. Over 90% of canine BUCs are invasive with metastatic potential [11,15].

The literature on canine BUCs concentrates mainly on the associated clinical practices and on the importance of a correct diagnosis, which could be the basis for prognostic follow-up studies. For this reason, canine BUCs are widely studied histologically, and multiple attempts have been made to propose grading systems over the decades.

The most common tumor variant in canine species is the papillary and infiltrating BUC. In these tumors, papillary or cauliflower-like structures projecting into the lumen are recognizable. These papillary projections show a central fibrous stalk, varying in thickness, covered by multiple layers of neoplastic urothelium that show mild-to-severe cellular atypia. Tumor cells can extend through the stalk of the tumor to the substantia propria or can reach the deeper muscle layers [16]. Moreover, tumor progression can be transmural, reaching the serosa. In advanced tumors, secondary projections or branching villous projections from the main tumor can emerge. When present, metastases, mainly associated with invasive BUCS, are generally located in the lungs, and are also frequent in the lymph nodes and bones [17,18].

The papillary and noninfiltrating BUC type has a similar luminal growth pattern but does not invade the stroma of the stalk or go beyond the lamina propria [16,17].

Nonpapillary and infiltrating BUCs are the second most common variant. These tumors appear as plaques, raised masses, or flat nodules. These tumors are often ulcerated and are more prone to infiltrating into the deeper muscle layers. The thickness of the bladder wall depends on the degree of invasion. These tumors are characterized by histological and cytological variability, and this BUC variant is the most likely to metastasize [14,16,17].

The least common variant is nonpapillary and noninfiltrating urothelial carcinoma, which is a flat lesion confined to the surface of the epithelium. It contains cells that are cytologically malignant and is considered synonymous with carcinoma in situ (CIS) [14,16,17].

Noteworthy, it is very important to distinguish BUCs from papillomas, which are defined as papillary tumors with a delicate fibrovascular stroma lined by less than seven layers of cytologically and architecturally normal urothelium, without increased cellularity or mitotic figures [13,14].

BUC neoplastic cells are polygonal with a variable amount of eosinophilic cytoplasm and sharp cell borders. The nuclei, round to oval, are generally large and vesicular, and nucleoli can be prominent. Varying degrees of differentiation and anaplasia can be present, and atypical nuclei and mitotic figures are common. Mitoses can be numerous, and bizarre mitotic figures can be seen. Within the tumor, areas of squamous and/or glandular metaplasia can be observed, but these should not change the diagnosis from the predominant cellular proliferation: urothelial epithelium. In cases of glandular metaplasia, cystic degeneration of the neoplastic epithelium mimicking the appearance of acini with lumina can be present [14,17].

BUCs are also characterized by the presence of large cytoplasmic vacuoles, called Melamed–Wolinska bodies, which may be empty or filled with homogeneous or stippled eosinophilic material. These structures are so characteristic of BUCs that, if seen in cytologic or histologic preparations from other locations, such as the lymph nodes, skin, or abdominal or pleural fluid, BUC should be listed as the most likely differential diagnosis [14,19].

Frequently, BUCs may stimulate a marked desmoplastic reaction in the primary and metastatic lesions [14]. This vigorous desmoplastic reaction does not seem to have a circumscribing function, but it has been indicated as a tumor-regulated response that may protect the neoplasm from the host’s cellular and soluble immune defenses [12,20]. Moreover, BUCs can be accompanied by various degrees of predominantly lymphoplasmacytic inflammation. The intensity of the inflammatory reaction seems to decrease as the depth of the invasion of the bladder wall increases, and a possible inverse correlation between metastases and lymphoid reaction was suggested by Valli et al. in 1995 and confirmed by Meuten in 2017 [12,14].

Valli et al. also described the way a carcinoma spreads into the wall of the bladder, which can be tentacular when the tumor infiltrates the strands, nests, and individual cells, inserting itself between normal bladder structures, or it can present a “broad front”, when the tumor advances uniformly though the bladder tissue [12]. Canine BUCs can invade adjacent tissues and organs, such as the ureter, prostatic urethra, and prostate gland, and moreover, the tumor can spread by means of vascular and lymphatic vessels [12,21,22]. Lymphatic invasion has recently been proven to be a significant negative prognostic factor [10]. However, metastases may occur with or without the observation of vascular invasion in the primary neoplasm [12,14].

## 3. Human BUCs: Histological Description

Histologically, human BUCs are classified as papillary or nonpapillary and infiltrating or noninfiltrating, based on the presence and extent of invasion, which is considered the most important element in pathologic evaluation. Contrary to what is reported in canine species, it is estimated that approximately 70–80% of human patients with newly diagnosed bladder cancer present with noninvasive or early invasive [23]. The histology of infiltrative BUCs shows infiltrating cohesive nests of cells with moderate-to-abundant amphophilic cytoplasm variably characterized by the presence of Melamed–Wolinska bodies and large hyperchromatic nuclei. The nucleus is typically pleomorphic and often has irregular contours with angular profiles. Nucleoli are highly variable in number and appearance, with some cells containing single or multiple small nucleoli and other cells having large eosinophilic nucleoli. The foci of marked pleomorphism may be seen, with bizarre and multinuclear neoplastic cells [24]. Mitotic figures are common, with numerous abnormal forms. The foci of squamous and glandular differentiation are common [25,26,27] and, occasionally, mucoid cytoplasmic inclusions may be present. In most cases, desmoplasia can be seen, and the stroma may contain a lymphocytic infiltrate with a variable amount of plasma cells. The inflammation may be mild to moderate and focal or severe, dense, and widespread. Intraepithelial neoplasia, including carcinoma in situ, is common in the adjacent urothelium [25].

Noninvasive BUCs are mainly characterized by papillary stalks that show frequent branching and form minimal-to-marked fusion. They show an orderly appearance with easily recognizable variations in architectural and cytologic features. Variations in nuclear polarity, size, shape, and chromatin pattern are frequently recognizable. The nuclei are uniformly enlarged with moderate-to-marked differences in shape, contour, and chromatin distribution. Nucleoli may be present, inconspicuous, or prominent. Mitoses are infrequent or common and occur basally or at any level [23].

## 4. Canine BUC Grading Systems

Some grading systems have been formulated for canine BUCs over the decades and, in general, they are based on human systems. The three main grading systems proposed for canine BUCs were proposed by Valli et al. in 1995, by Patrick et al. in 2006, and, the most recent, by Meuten in 2017. These systems, even though similar to those used in human pathology, are still neither routinely applied nor validated with prognostic studies in veterinary medicine [12,13,14]. 

The first grading system for canine BUCs described in the literature was proposed by Valli et al. in 1995, taking the 1973 human grading system as a reference [12,21]. Accordingly, Valli subdivided the BUCs into three grades, and in situ lesions were classified as grade 1 in most cases. This distinction in grades is mainly based on nuclear appearance, including nuclear position, shape, the appearance of the chromatin, and the presence of prominent nucleoli (Table 1).

With the system of Valli et al., which includes three grades as per the WHO’s 1973 human grading system, several tumors are classified as grade 2, having intermediate morphologic characteristics. Moreover, Valli et al. considered as discriminating criterion only the nuclear morphology, while other possible morphologically relevant features, such as tumor architecture or invasion, are not mentioned.

A second veterinary grading system, strictly limited to papillary BUCs, was proposed by Patrick et al. in 2006, based on the WHO’s human BUC grading of 2004. In the grading system of Patrick et al., as per Valli et al., the classification criteria are the same as those used in human medicine and are reported in Table 2. [13]. Patrick et al. mentioned the infiltrating behavior of the tumor; however, it is not clear whether the depth of infiltration can be a discriminating criterion for assigning the histological grade.

Recently, in 2017, based on the system described by Cheng in 2012 for human BUCs, a new grading system for canines was proposed (Table 3) [14,28]. To assess the histological grade, this system focuses on the invasiveness of the tumor: Invasive tumors are considered high grade, while noninvasive tumors are classified as low grade. Moreover, as in other grading systems, cell polarity, tumor architecture, mitotic activity, and nuclear atypia are also considered.

Although this grading system was proposed based on the system of Cheng et al. from 2012, it includes only two grades, resembling the human one proposed by the WHO in 2004. However, unlike the WHO’s 2004 system, in this recent veterinary grading system, the tumor invasiveness is clearly stated as a discriminating feature for high-grade BUCs.

## 5. Human Grading Systems

The histopathologic grade of a BUC is considered, in human medicine, as one of the best predictors of its biologic behavior. However, the criteria for the pathologic grading of BUCs have been a source of controversy for many decades [21,28,29].

In human medicine, the classification and grading for BUCs have been debated since the introduction of a three-tiered grading system for noninvasive papillary urothelial neoplasms (1973 WHO grades 1–3) [21]. The histological criteria adopted in the 1973 BUC grading system include tumor architecture, cellular features, nuclear/cytoplasmic ratio, mitosis, and nuclear characteristics. Although the grading system appears to refer only to noninfiltrating tumors, invasiveness is considered and mentioned as a possible tumor characteristic in the various grades (Table 4) but, contrary to the grading systems proposed more recently, not listed as a discriminating criterion. The histological criteria of the 1973 grading system are schematized in Table 4 [30].

Of note, when a histological grading system that includes three grades is applied to tumors, a great number of samples are considered as intermediate-grade tumors. Nevertheless, this wide intermediate-grade group could include BUCs characterized by very different morphological features. This is quite expected, since all the biological variables tend to be statistically distributed with a normal distribution in which the intermediate samples represent 95% and the extremes represent just 5% of the samples [31].

In 1998, a revised system for classifying papillary urothelial neoplasms of the urinary bladder was proposed [28]. This system was subsequently formally adopted by the WHO (1998 WHO/ International Society of Urological Pathology [ISUP] classification). In 2004, a further classification system for noninvasive papillary urothelial neoplasms, a slight modification of the 1998 WHO/ISUP classification, was published in Pathology and Genetics of Tumors of the Urinary System and Male Genital Organs, one of the WHO’s “Blue Books” for the classification of tumors [23,28]. This new system separates noninvasive papillary urothelial neoplasms into four categories: Two categories are referred to as benign lesions, such as papilloma and urothelial proliferation of uncertain malignant potential (PUNLMP). The other two categories are concerned with malignant lesions and include low- and high-grade carcinomas (Table 5). The term urothelial proliferation of uncertain malignant potential (PUNLMP) has been introduced, supplanting the term hyperplasia [32,33]. It describes a thickened urothelium with minimal or no cytological atypia and no true papillary fronds, although undulations of the surface epithelium can be common [28,34]. On the contrary, invasive BUCs are classified separately. Invasive BUCs have a propensity for divergent differentiation, with the most common being squamous, followed by glandular and undifferentiated carcinomas, nested variants, microcystic variants, micropapillary variants, lymphoepithelioma-like carcinomas, lymphoma-like, plasmacytoid variants, sarcomatoid, BUC with giant cells, BUC with trophoblastic differentiation, clear cell variants, and lipid cell variants [23].

It is perceivable that a grading system with two malignant grades could be easier and faster to apply in routine diagnostics. However, it does not reflect the biological variability of tumors, which includes intermediate morphological features that, not being considered in this tumor grading, could create a difficulty for the pathologist, forcing a grade assignment. In the following decade, in 2012, Cheng provided a comparative and critical view of the grading systems in use in human medicine and proposed a new classification for BUCs [28] that considered two low grades and two high grades (Table 6). In this system, the diagnostic criteria for urothelial papilloma are identical to those defined by both the 1973 and the 2004 WHO grading systems, while Grade 1 (low-grade) tumors are classified as PUNLMP in the WHO’s 2004 classification system. Moreover, Cheng adopted a sharp schematic approach that included numerous histologic and cellular criteria, such as increased cell layers, superficial umbrella cell presence, polarity/overall architecture, discohesiveness, clear cytoplasm, nuclear size, nuclear pleomorphism, nuclear polarization, nuclear hyperchromasia, nuclear grooves, nucleoli, and mitotic figures. In addition, stromal invasion is also mentioned, particularly for high-grade BUCs, even if not considered as a discriminative feature for grading assignment. In Table 6, the description of the histological grade proposed by Cheng et al. (2012) (limited to malignant tumors) is schematized.

The presence of stromal and deep-layer invasion, which seems to be a key feature, is considered in a more recent BUC grading system formulated by the WHO in 2016 [34], which, based on the previous one edited in 2004 [23], has refined discriminating criteria (Table 7). Apart from benign lesions, the WHO’s 2016 grading system includes two grades of BUCs, low grade and high grade, which can be invasive or not. Although a small percentage of invasive BUCs are low grade, usually limited to the lamina propria, more than 95% of invasive tumors are high grade [34]. However, all detrusor muscle–invasive BUCs are directly assigned to the high grade. Moreover, tumors that overcome the basement membrane and extend to the lamina propria, even if not invading the muscle layer, can be accompanied by lymphovascular invasion and metastatic spread. As such, in many instances, pathologists identify these tumors as high grade, independently of their cellular atypia.

As already highlighted, this new grading system considers, for the first time, the infiltrative behavior of BUCs as and first mentions tumors extending through the chorion and their possibility to also invade the blood and lymphatic vessels. These invasive tumors are assigned to the high grade independently from their histological atypia. This is important because, for the first time, tumor infiltration is considered as a discriminating feature, which could immediately lead a pathologist to classify a tumor as high grade.

## 6. Critical Evaluation

In veterinary pathology, there are several histopathological grading systems, according to the tumor type. A review of these systems was recently published emphasizing those already widely and successfully applied in veterinary diagnostic routine [1]. Among these are Peña’s grading system for mammary tumours [35] and those formulated by Patnaik and Kiupel for mast cell tumors [36,37]. Concerning BUCs, only Meuten’s (2017) system was included in the review [1]; however, it was clearly pointed out that for canine BUCs, specific features or cutoffs (e.g., mitotic count) are not available, and studies validating BUC grading systems by the assessment of their prognostic relevance are lacking [1].

The lack of studies is probably due to the fact that none of the grading systems proposed for canine BUCs are commonly used in diagnostic practice. This could be due mainly to a lack of extensive follow-up studies testing the reliability of these grading systems and to the late stage at the time of diagnosis in the vast majority of dogs. On the contrary, the systems described in the literature for canine BUCs could be difficult to apply and/or could lead to interobserver variability. Another important feature is muscular invasion as a criterion of malignancy; however, some samples, especially from biopsy, do not include all of the urinary bladder layers to allow for this evaluation.

Moreover, the lack of follow-up studies demonstrating the association between the grading systems and clinical outcomes discourages clinicians from performing a biopsy for tumor classification. In clinical routines, catheter-based cytology and molecular diagnosis (evaluation of *BRAF* mutation) have been used for final diagnosis, and in advanced tumors, biopsy is not usually performed for the application of grading systems [22].

The canine BUC grading systems proposed by Valli et al. (1995) and Patrick (2006) [12,13] show various limitations, such as divisions into three grades with no clear cut-offs between them, which leads pathologists to lump tumors on the edges of Grades 1 and 3 into Grade 2. Consequently, this results in a wide range of reported incidences for Grade 2 BUCs, as observed in human [28] and veterinary medicine [14]. Moreover, a lack of a precise cut-off values and clearly defined distinguishing features between grades leads to inter- and intra-pathologist variation [14].

To bypass these concerns, grading systems organized into only two grades were further proposed in human and veterinary medicine [14,23,34]. However, these systems could force pathologists to include a tumor as either low or high grade, with the possibility of under- or overestimating existing intermediate tumors. On the contrary, the two-grade 2016 human system by the WHO and the similar veterinary one (proposed by Meuten in 2017) positively introduced invasiveness as an important discriminating feature, while Meuten’s system includes invasive BUCs directly as high grade.

This is an important contribution since, for the first time in veterinary medicine, tumor invasiveness is mentioned as a discriminating criterion and invasive BUCs are included in the high grade based on evidence that approximately 90% of them are diagnosed at the advanced stage. Nevertheless, the increased attention paid by dog owners, together with the new diagnostic techniques, could lead to an earlier detection of the tumor in the near future. In this scenario, and for future studies, it would be important to record the eventual invasiveness of the tumor, such as whether it is minimally or highly invasive, since different grades of infiltration could correspond to a different prognosis. 

Moreover, such a scenario suggests that, in future follow-up studies on canine BUCs, more than one grading system could be applied to find the most reliable system with the highest prognostic value. Among the possible grading systems to test for canine BUCS, the human one proposed by Cheng, articulated in four grades, and its simplified version that includes only the two grades proposed by Meuten, may be used because, in light of the above, one or both of them could be the most reliable in veterinary medicine. 

Moreover, histologic classifications cannot serve as meaningful surrogate endpoints for prognostic studies, but they may serve as a basis for building hypotheses, considering their relationship with the predictive markers that are associated with a treatment outcome [38,39].

In addition, research studies have also demonstrated that vascular invasion correlates with a worse outcome in human patients [40], and lymphatic invasion was also recently associated with a worse prognosis in canine BUCs [22]. Lymphovascular (LVI) invasion is recognized as a marker of tumor malignancy, suggesting aggressive biological behavior and increased probability of metastatic disease. However, when formulating a diagnosis criterion used to distinguish LVI from pseudo-vascular invasion and retraction, artifacts must be also considered. These criteria are thrombus adherence to intravascular tumor, tumor cell invasiveness through a vessel wall and endothelium, and the presence of neoplastic cells within a space lined by lymphatic or blood vascular endothelium; therefore, the presence of neoplastic cells in lymphatic or blood vessel should be confirmed by immunohistochemistry [41]. The first two criteria are considered strict and may be more likely to predict metastases than the latter two and must be included in the diagnostic report [42].

For these reasons, if present, we would recommend including vascular and lymphatic invasion in diagnostic reports, independently from the grading system used and from the tumor grade assigned.

Retrospective and prospective follow-up studies are needed in veterinary medicine to determine which histological grading could have the most accurate prognostic value for patients or to eventually propose a new suitable grading system to better characterize canine BUCs.

## Figures and Tables

**Table 1 animals-12-01455-t001:** Scheme of canine BUC grading (Valli et al., 1995) [12].

GRADE 1	Well-differentiated BUCs. Normal cytoplasmic volume and regularity of nuclear placement. Nuclei round with mild anisokaryosis and hyperchromicity. Nucleoli are small or inapparent.
GRADE 2	Moderately differentiated. Moderate variations in cytoplasmic volume and in nuclear placement, size, and shape. Nuclei are hyperchromatic, most having a prominent single nucleolus.
GRADE 3	Anaplastic. Marked variation in cell and nuclear size and shape with irregular nuclear crowding and molding. Chromatin deeply stained and irregularly distributed. Nucleoli are prominent, frequent, and variable in location.

**Table 2 animals-12-01455-t002:** Scheme of the canine BUC grading proposed by Patrick et al. in 2006 [13].

GRADE 1 (Low Grade) Papillary UC	Overall orderly appearance and easily recognizable variations in architectural and/or cytologic features. Nuclear size and shape, and chromatin texture vary. Frequent mitotic figures usually seen in the basilar half. Adjacent papillae may be fused.
GRADE 2 (High Grade) Papillary UC	Overall disorderly appearance, but some degree of polarity is retained. Cells irregularly clustered, epithelium is disorganized. Cytological moderate anaplasia. Clumped nuclear chromatin, nucleoli may be prominent. Mitotic figures, including atypical forms, may be seen at all levels. May invade the lamina propria or muscularis propria.
GRADE 3 (High Grade) Papillary UC	Completely disordered appearance and lack of polarity. Cells irregularly clustered and epithelium is disorganized. Cytologically marked pleomorphism. Clumped nuclear chromatin, nucleoli may be prominent. Mitotic figures, including atypical forms, may be seen at all levels. Invasion of the lamina propria or muscularis propria may be present.

**Table 3 animals-12-01455-t003:** Scheme of the canine grading system proposed by Meuten in 2017 [14].

Low Grade	Papillae or flat, orderly cell to cell. Mild-to-moderate cellular atypia. Nuclear abnormalities present: enlarged nuclei, nucleoli visible, with limited to no mitoses. No invasion.
High Grade	Papillae or flat, loss of cell polarity, disorganized growth. Marked cellular atypia. Marked nuclear pleomorphism: chromatin clumped, nucleoli prominent. Mitoses numerous, some abnormal. Invasion present, state depth of infiltration, and if UC in blood vessels or lymphatics.

**Table 4 animals-12-01455-t004:** Scheme of the WHO’s 1973 grading system for human BUCs.

Benign Tumors	
Papilloma	Papillary tumor with a delicate fibrovascular stroma covered by a regular transitional epithelium indistinguishable from that of the normal bladder and not more than six layers thick.
Malignant Tumors	
GRADE 1	Almost always noninvasive, consisting of a thin fibrovascular core covered by a thickened transitional cell epithelium (more than six cells thick), exhibiting only slight architectural and cellular abnormalities. Minimal nuclear pleomorphism, nuclear cytoplasmic ratio increased without prominence of a nuclear membrane or chromatin. Mitoses uncommon or present in the basal and intermediate cell layers.
GRADE 2	Most commonly noninvasive. Papillae shorter and blunter than in Grade 1, lesions with a wider fibrovascular core. Moderate loss of base-to-surface differentiation in the epithelium and usually a disturbance in cellular polarity. Pleomorphic and/or large nuclei. Mitotic figures’ common nuclear cytoplasmic ratio increased, nuclear membrane thickened, and clumped chromatin.
GRADE 3	Frequently invasive. Papillary projections of the neoplastic epithelium with no differentiation from base to surface, marked nuclear pleomorphisms, and high nuclear cytoplasmic ratio. Mitoses frequent and bizarre.

**Table 5 animals-12-01455-t005:** Scheme of the WHO’s 2004 grading system, excluding benign tumors for human urothelial tumors.

Low Grade	Slender, papillary stalks with branching and minimal fusion. Orderly appearance with easily recognizable variations in architectural and cytologic features. Variations in nuclear polarity, size, shape, and chromatin pattern. Nuclei uniformly enlarged with mild differences in shape, contour, and chromatin distribution. Nucleoli present but inconspicuous. Mitoses infrequent at any level, more frequent basally.
High Grade	Papillary architecture with fused papillae and branching. Predominant pattern of disorder with easily recognizable variations in architectural and cytologic features. Marked variations in nuclear polarity, size, shape, and chromatin pattern. Nuclei often pleomorphic with moderate-to-marked variation in size and irregular chromatin distribution. Nucleoli are prominent. Frequent mitoses that may be atypical and occur at any level. The thickness of the urothelium may vary, often with cell dyscohesion. May be invasive.

**Table 6 animals-12-01455-t006:** Scheme of Cheng et al. from 2012, a grading system for malignant human urothelial carcinoma.

Characteristics	Grade 1 (Low Grade)	Grade 2 (Low Grade)	Grade 3 (High Grade)	Grade 4 (High Grade)
Increased cell layer (>7)	Yes	Variable	Variable	Variable, usually <7 layers
Superficial umbrella cells	Present	Often present	Usually absent	Usually absent
Polarity/overall architecture	Normal	Mildly distorted	Moderately distorted	Severely distorted
Discohesiveness	Normal	Normal	Mild to moderate	Severe
Clear cytoplasm	May be present	May be present	Usually absent	Usually absent
Nuclear size	Normal or slightly increased	Mildly increased	Moderately increased	Markedly increased
Nuclear pleomorphism	Uniform, slightly elongated to oval	Mild, round to oval with slight variation in shape and contour	Moderate	Marked
Nuclear polarization	Normal to slightly abnormal	Abnormal	Abnormal	Absent
Nuclear hyperchromasia	Slight or minimal	Mild	Moderate	Severe
Nuclear grooves	Present	Present	Absent	Absent
Nucleoli	Absent or inconspicuous	Inconspicuous	Enlarged, often prominent	Multiple, prominent
Mitotic figures	None/rare, basal location	May be present at any level	Often present	Prominent and frequent, atypical forms
Stromal invasion	Rare	Uncommon	May be present	Often present

**Table 7 animals-12-01455-t007:** Scheme of the grading system by the WHO in 2016, excluding benign tumors for human BUCs.

Low Grade	Orderly arranged papillae. Variations in polarity and nuclear size, shape, and chromatin distribution not of primary importance. Specific cytological disorder exists. Rare mitosis, if present, usually occurs in the lower half of the urothelium.
High Grade	Completely disorderly appearance due to both cytonuclear and architectural disorganization. Wide spectrum of pleomorphism from moderate to marked. Nuclei pleomorphic with prominent nucleoli, frequent mitosis. Intraepithelial necrosis may be present. Variable thickness papillae are fused displaying anarchic growth. Considered aggressive lesions. Can be infiltrating or not.

## Data Availability

Not Applicable.

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
