# Peer review of "Grading Systems for Canine Urothelial Carcinoma of the Bladder: A Comparative Overview"

_animals, 2022, doi:10.3390/ani12111455_

Round 1

Reviewer 1 Report

Simple Summary

Line 16: the word 'favoring' does not seem most suitable. Consider replacing by 'facilitating', 'supporting' or similar

Line 18: rephrase as: … already have a histological grading system used for these purposes, but some of these schemes…

Line 23: replace 'either' by 'neither'

Abstract

Line 38: replace 'have proposed' by 'have been proposed'

Introduction

Line 82: indicated reference 10 does not cover morphology and metastasis location. Indicate an alternative reference.

Line 88: rather than stating 'unclear', I would recommend indicating some of the most likely reasons. Consider i) unknown relevance for prognosis and therapy, ii) late stage disease at the time of diagnosis in the vast majority of dogs, iii) limited acceptance among pathologists to adopt new grading system among others

Canine BUCs: Histological description

Line 101: I recommend adding two more features: tumor differentiation (urothelial vs. divergent) and tumor stroma (including presence and extent of inflammation)

Line 102: add a more recent reference

Line 105: replace 'chorion' by 'lamina propria'

Line 111: does 'morphologically' refer to the microscopic morphology? If yes, it is the same as 'histologically' in this context. Rephrase.

Line 129: replace 'They' by 'These tumors' (or similar)

Line 130: replace 'this last variant' by 'this variant' or by 'this BUC variant'

Line 142: syncytial cells do not seem to be that common (?). If yes, please add a suitable reference confirming this

Line 152: replace 'they' by 'BUC'

Line 158: plasma cells are also frequently present. Consider to replace 'various degrees of lymphoid' by 'various degrees of predominantly lymphoplasmacytic'

Lin 161: add a more recent reference

Human BUCs: Histological description

  • Avoid statements about canine BUC in this section (lines 174-177)
  • Add few sentences about features mentioned under 'canine BUCs', such as Melamed-Wolinska bodies and desmoplasia

Lines 183-189: reorder sentences to avoid mixing of tumor cellular features, stroma…etc. Recommended order: 1) Mitotic figures…, 2) The foci of squamous…, 3) Occasionally, mucoid…, 4) In most cases, …. The inflammation may…, 5) Intraepithelial neoplasia, …

Line 195: replace 'could' by 'are'

Line 196: delete 'could'

Line 208: I recommend to delete 'color' (nuclear 'color' i.e. basophilia is given by the chromatin density/pattern which is already mentioned as feature)

Human grading systems

  • It would be relevant for the readers of this review to know the current guidelines/recommendations regarding the choice and application of the grading system (which system is to be used? Is more than one system accepted? Who makes these recommendations (WHO or other)?
  • Clearly indicate for each different system if they refer to invasive or non-invasive BUC

Line 271: To my understanding, squamous, glandular and undifferentiated carcinomas are separate entities and not BUC subclasses. Rephrase.

Critical evaluation

  • It would be interesting for the readers of this review if there are any lessons to be learnt from other tumor types (where a grading system was successfully established and which is being used in routine veterinary diagnostics). I recommend making a comparison with another canine tumor type.
  • It would improve the manuscript if you add a paragraph about the validation of the different BUC grading systems in dogs. Which system was validated? How and when and with which conclusion?

Line 363-365: Indicate that HE stain may have limitations for the assessment of vascular invasion (especially so due to the common tissue retraction artefacts in BUC). This may be one reason why this feature may not be included in a grading system that is based on HE only.

References

  • Several references with missing colon after the year of publication
  • Update the most recent references (published i.e. printed version rather than the online one)

Author Response

REVIEWER 1

Simple Summary

Line 16: the word 'favoring' does not seem most suitable. Consider replacing by 'facilitating', 'supporting' or similar. Replaced “facilitating with “supporting” according to your suggestions. Line: 16.

Line 18: rephrase as: … already have a histological grading system used for these purposes, but some of these schemes…Rephrased according to your suggestion line 17-18.

Line 23: replace 'either' by 'neither'. Corrected according to your suggestion. Line: 23

Abstract

Line 38: replace 'have proposed' by 'have been proposed'. Corrected accordingly to your suggestion. Line: 40.

Introduction

Line 82: indicated reference 10 does not cover morphology and metastasis location. Indicate an alternative reference. Corrected according to your suggestion. Line: 83.

Line 88: rather than stating 'unclear', I would recommend indicating some of the most likely reasons. Consider i) unknown relevance for prognosis and therapy, ii) late stage disease at the time of diagnosis in the vast majority of dogs, iii) limited acceptance among pathologists to adopt new grading system among others. Corrected according to your suggestions. Lines: 89-93.

Canine BUCs: Histological description

Line 101: I recommend adding two more features: tumor differentiation (urothelial vs. divergent) and tumor stroma (including presence and extent of inflammation). Added according to your suggestion. Line: 103-105.

Line 102: add a more recent reference. Added according to your suggestion. Line: 105.

Line 105: replace 'chorion' by 'lamina propria'. Replaced accordingly to your suggestion. Lines: 108-109.

Line 111: does 'morphologically' refer to the microscopic morphology? If yes, it is the same as 'histologically' in this context. Rephrase. Yes it is. Rephrased according to your suggestion. Line 117.

Line 129: replace 'They' by 'These tumors' (or similar). Corrected according to your suggestion. Lines: 131-132, 134.

Line 130: replace 'this last variant' by 'this variant' or by 'this BUC variant'. Replaced according to your suggestion. Line: 135.

Line 142: syncytial cells do not seem to be that common (?). If yes, please add a suitable reference confirming this. Deleted accordingly to your suggestion. Line: 147.

Line 152: replace 'they' by 'BUC'. Corrected according to your suggestion. Line: 158.

Line 158: plasma cells are also frequently present. Consider to replace 'various degrees of lymphoid' by 'various degrees of predominantly lymphoplasmacytic'. Corrected according to your suggestion. Line: 164.

Line 161: add a more recent reference. Added according to your suggestion. Lines: 167-168.

Human BUCs: Histological description

  • Avoid statements about canine BUC in this section (lines 174-177). Delated according to your suggestion. Line: 184.
  • Add few sentences about features mentioned under 'canine BUCs', such as Melamed-Wolinska bodies and desmoplasia. Added according to your suggestion. Line: 186.

Lines 183-189: reorder sentences to avoid mixing of tumor cellular features, stroma…etc. Recommended order: 1) Mitotic figures…, 2) The foci of squamous…, 3) Occasionally, mucoid…, 4) In most cases, …. The inflammation may…, 5) Intraepithelial neoplasia, …. Reordered according to your suggestion. Lines: 192-194.

Line 195: replace 'could' by 'are'. Replaced according to your suggestion. Line: 203.

Line 196: delete 'could'. Delated according to your suggestion. Line: 204.

Line 208: I recommend to delete 'color' (nuclear 'color' i.e. basophilia is given by the chromatin density/pattern which is already mentioned as feature). Delated according to your suggestion. Line: 216.

Human grading systems

  • It would be relevant for the readers of this review to know the current guidelines/recommendations regarding the choice and application of the grading system (which system is to be used? Is more than one system accepted? Who makes these recommendations (WHO or other)?

Thank you for the question, that’s the problem! Guidelines are different for each hospital and country. Here in Italy guidelines suggest the use of both the WHO 1973 and the WHO 2004 grading systems.

  • Clearly indicate for each different system if they refer to invasive or non-invasive BUC. Grading systems are used in human medicine only on non-infiltrating tumors because infiltrating ones are considered apart. In veterinary this feature is not specified. Only Meuten mentioned the infiltration considering this aspect to distinguish a non-infiltrating tumor which is considered low grade from an infiltrating one that is considered automatically as a high grade tumor. See line: 343.

Line 271: To my understanding, squamous, glandular and undifferentiated carcinomas are separate entities and not BUC subclasses. Rephrase. Rephrased according to your suggestion. Lines: 278-283.

Critical evaluation

  • It would be interesting for the readers of this review if there are any lessons to be learnt from other tumor types (where a grading system was successfully established and which is being used in routine veterinary diagnostics). I recommend making a comparison with another canine tumor type. Added according to your suggestion. Lines: 328-331.
  • It would improve the manuscript if you add a paragraph about the validation of the different BUC grading systems in dogs. Which system was validated? How and when and with which conclusion?

Thank you for the question. All the grading systems presented for canine BUCs are not in use in common routine diagnosis since they are not already validated. Contrary to other tumor type such as mast cell tumor or tumors of the mammary gland, literature lack of follow-up studies in which the correlation between the grade of the canine BUC and patient outcome is investigated. Added lines 227-235.

Line 363-365: Indicate that HE stain may have limitations for the assessment of vascular invasion (especially so due to the common tissue retraction artefacts in BUC). This may be one reason why this feature may not be included in a grading system that is based on HE only.

Thank you for the question. Criteria used to distinguish limphovascularinvasion from pseudo-vascular invasion and retraction artifacts have been added. Lines: 385-395.

References

  • Several references with missing colon after the year of publication. Corrected according to your suggestion.
  • Update the most recent references (published i.e. printed version rather than the online one). Corrected according to your suggestion.

Reviewer 2 Report

very good review

too long, up to authors and editor if you wish to shorten

add a few refs especially webster and read this excellent ms and add some concepts

consider prognostic and predictive parameters

you need to add % of human and canine UCC that are aggressive vs indolent

Author Response

Dear reviewer, thanks for the interesting suggestions. Here what has been corrected basing on your suggestions.

 UCC GRADING – review

Line 29 – consider adding ( predictive markers) …. personalized treatment ( predictive markers). Added according to your suggestion. Lines: 30-31.

You should add the philosophy / principle of predictive markers to your ms that emphasize prognostic markers. Vet path favors development of prognostic markers = those that predict an outcome – mets, recurrence, survival metrics etc but the same markers used to help provide a prognosis can be trialed as “predictive markers” = help select a treatment….. if prospective studies will evaluate this, and if we emphasize their importance in review ms such as yours.

I believe you are trying to develop a grading system for UCC by merging existing systems, selecting good features in each one – KEY – is to THEN correlate parameters chosen with standardized outcomes. “All” grading systems are worthless if they do not predict something = an outcome or help choose a treatment. Please emphasize this. Lines 31 – 33 We must have accurate standardized outcomes (e.g. RECIST) and outcome data must be collected as carefully as is the pathology portion of the study. Therefore, encourage prospective studies to include different treatments and determine if some of the parameters can be used for prognoses and some can be used to select treatment = predictive marker.s See Webster I do not believe you cited this reference and it should be. Please read Webster ref – it is excellent and you will likely include some of the principles the authors described.

Webster, J., M. M. Dennis, N. Dervisis, J. Heller, N. J. Bacon, P. J. Bergman, D. Bienzle, G. Cassali, M. Castagnaro and J. Cullen (2011). "Recommended guidelines for the conduct and evaluation of prognostic studies in veterinary oncology." Veterinary pathology 48(1): 7-18.

Thank you for your interesting suggestions we have transposed and reworked your suggestions and added a sentence. Lines: 34-35.

Line 29 personalized treatment for each patient you could cite ms on Precision Oncology = same principle, just different words. Thank you for the suggestion, however, if possible, we prefer to maintain “personalized treatments for each patients” considering these more suitable at the moment. The precision oncology is an interesting field only recently approached in human medicine for bladder cancer and still not investigated on canine BUC. We will consider this fields in future studies.  

Line 39 = the reason for their scarce application is mainly related to a 39 lack of specific cutoffs = another reason is UCC in dogs do not have a range of behavior, as they do in humans. Canine UCC are almost always aggressive, indolent forms are rare or rarely seen. Here or someplace you will need to point that out. A weakness of canine BUC as a model is – they have few indolent variations as compared to humans – please point this out and discuss.

Thank you for the question and for suggesting reliable motivation. The scarce application of canine grading systems in mainly due to the lack of follow up studies as reported now in line 343 and to the lack of cut-offs which may be useful to clearly distinguish a tumor grade from the others. Added  comments at lines: 89-92.

52 – and predicting treatments - please keep prognostic and predictive throughout. Corrected according to your suggestion. Line: 53.

84 = present and consolidated what does this mean? Reword? Rephrased according to your suggestion. Lines 328-331.

91 – delete. Delated according to your suggestion.  Line: 93.

190s- tell us approximately how many are indolent (2/3) vs aggressive (1/3) – differences and similarities with canine BUC should be included. Added according to your suggestion. Lines: 112-114 and 182-184.

323s- point out the canine BUC, to date do not have a range of behaviors or something like – we do not see them until the tumor is aggressive; perhaps because the tumor remains occult in vet med? You do so around lines 350 – please emhpaszie. Added and stressed out according to your suggestion, lines: 90-91, 338-339.

360s= regard vascular and lymphatic invasion – see vcgp.org on this topic and mention that hard criterion should be used to differentiate pseudoinvasion vs true 2. Added according to your suggestion. Lines: 384-395.

CONSIDER – grading systems are based on histology, however, investigations that correlate a newly proposed system with outcomes should broaden the scope of the characteristics evaluated and try to incorporate cytology (common dx that will only increase in use; and very useful for BUC) AND molecular profiles of tumor and host. Do not miss the opportunity to gather additional data. The latter is important in predicting behaviors and selecting treatments for human cancers.............yada yada mention we should be reaching beyond histology as we strive to improve the discriminating ability of grading systems based on histology.............. when gathering data for one purpose , collect data for other means. if we do not do this we will or may miss opportunities to identify discriminating features

Really thank you for your comments we added a sentence according to your suggestion. Lines: 377-380.